# Infrared Small Target Detection via Non-Convex Rank Approximation Minimization Joint $l_{2,1}$ Norm

**Landan Zhang, Lingbing Peng, Tianfang Zhang, Siying Cao and Zhenming Peng *** 

School of Information and Communication Engineering, University of Electronic Science and Technology of China, Chengdu 610054, China; zhanglandan@std.uestc.edu.cn (L.Z.); lbpeng163@163.com (L.P.); sparkcarleton@gmail.com (T.Z.); caosiying3008@163.com (S.C.)

*** Correspondence: zmpeng@uestc.edu.cn; Tel.: +86-130-7603-6761

**Abstract:** To improve the detection ability of infrared small targets in complex backgrounds, a novel method based on non-convex rank approximation minimization joint $l_{2,1}$ norm (NRAM) was proposed. Due to the defects of the nuclear norm and $l_1$ norm, the state-of-the-art infrared image-patch (IPI) model usually leaves background residuals in the target image. To fix this problem, a non-convex, tighter rank surrogate and weighted $l_1$ norm are instead utilized, which can suppress the background better while preserving the target efficiently. Considering that many state-of-the-art methods are still unable to fully suppress sparse strong edges, the structured $l_{2,1}$ norm was introduced to wipe out the strong residuals. Furthermore, with the help of exploiting the structured norm and tighter rank surrogate, the proposed model was more robust when facing various complex or blurry scenes. To solve this non-convex model, an efficient optimization algorithm based on alternating direction method of multipliers (ADMM) plus difference of convex (DC) programming was designed. Extensive experimental results illustrate that the proposed method not only shows superiority in background suppression and target enhancement, but also reduces the computational complexity compared with other baselines.

**Keywords:** infrared image; small target detection; non-convex rank approximation minimization; structured norm

## 1. Introduction

Infrared search and track (IRST) systems have been widely used in many applications, especially in the military field. As a basic function of IRST systems, infrared small target detection plays an important role in early-warning, precision guidance, and long-range target detection. However, the lack of texture and structure information of the target under infrared conditions, coupled with the influence of long distance, complex background, and various clutter, means that the infrared small target is often spot-like, even submerged in the background, which makes it extremely difficult to detect. Therefore, it is a challenge to design a detection method that is effective and adaptable to various complex backgrounds.

Generally speaking, infrared small target detection methods can be divided into two categories: sequential-based and single-frame-based methods. Sequential-based methods need to join multiple frames to capture the target trajectory and avoid noise interference. When the background is homogeneous and the target moves slowly, sequential-based methods such as the Robinson Guard spatial filter [1], dynamic programming algorithm [2], 3D matched filtering [3], and so on can perform well. Nevertheless, in real applications, not only is it difficult to guarantee the homogeneity of the background, but the movement between the target and imaging sensor is fast, resulting in the rapid degrading of sequential-based methods. Meanwhile, high storage and hardware requirements make

this approach unsuitable for large-scale engineering projects. Although there are still some studies on sequential methods [4,5], single-frame based methods are more popular and have attracted more research attention in recent years [6–8].

Conventional single-frame based detection approaches, like Tophat filter [9], Maxmean and Maxmedian filter [10] utilize filtering to suppress the background and enhance the target under the assumption of background consistency. These methods can achieve good performance with a uniform background and are easily implemented. Unfortunately, most backgrounds do not meet the ideal assumption, leading to a high false alarm ratio. Recently, approaches based on human visual system (HVS) [11] have been developed to improve the performance of infrared small target detection with the assumption that the target is the most salient object, most of which compute the saliency map in a manner of filtering [7,12–18] or transforming [19,20]. Actually, in many real scenes, the most prominent object is not the desired target while the highlight edges and salient non-target components can destroy the whole detection. Both of these methods are sensitive to noise and do not perform well when the size of the target varies within a large range.

Conventional detection methods based on simple assumptions merely consider either the background or the target. In order to overcome the limitations of the traditional methods, a state-of-the-art approach based on robust principal component analysis (RPCA) called the infrared patch-image (IPI) model was proposed [21]. Focusing on the non-local self-correlation configuration of the background, the assumption of the IPI model fits real scenes well. Unlike the general infrared image model, the IPI model reformulates it as follows:

$$D = B + T + N \tag{1}$$

where $D$, $B$, $T$, and $N$ are the corresponding patch-images of the original infrared image, the background image, the target image, and the random noise image, respectively. Then, with the low-rank property of $B$ and the sparsity of $T$, the conventional small target detection problem is converted to a RPCA optimization problem (RPCA is also called sparse and low rank matrix decomposition) as follows:

$$\begin{aligned} \min_{B,T} & \|B\|_* + \lambda \|T\|_1 \\ \text{s.t.} \quad & D = B + T \end{aligned} \tag{2}$$

where $\|\bullet\|_*$ denotes the nuclear norm of a matrix (i.e., the sum of singular values), $\|\bullet\|_1$ is the $l_1$ norm (i.e., $\|X\|_1 = \sum_{ij} |X_{ij}|$), and $\lambda$ is a positive weighting constant.

Convex problem (2) is efficiently solved in Reference [21] via the accelerated proximal gradient (APG) [22] approach. However, since nuclear norm minimization (NNM) treats all singular values equally, it may obtain suboptimal solutions [23]. The $l_1$ norm faces the same problem, which means that Equation (2) cannot recover the background and target image precisely. In other words, when facing complex infrared scenes, the performance of IPI is weak, namely, strong edge residuals remain in the recovered target image and the recovered background is blurry (poor edge retention). To approximate the rank more accurately, many non-convex surrogates have been introduced to RPCA one after another because of their flexibility including weighted nuclear norm minimization (WNNM) [24], truncated nuclear norm minimization (TNNM) [25], capped norm [26], the Schatten-p norm [27] and so forth. Similarly, weighted $l_1$ norm [28], capped $l_1$ norm [29], and $l_{2,1}$ norm [30] have also been proposed to improve the ability to represent sparsity.

Furthermore, alternating direction method of multipliers (ADMM) [31] has a faster convergence rate when compared with APG and its precision is also higher [32]. Hence, ADMM is a popular method to solve optimization problems in many fields, particularly in the computer vision and image processing field. For instance, Gu et al. [33] utilized ADMM to solve the non-convex weighted nuclear norm minimization (WNNM) problem efficiently, which was successfully applied to image inpainting. Xue et al. [34] proposed a non-convex low-rank tensor completion model using ADMM to obtain the best recovery result in color images. Based on total variation regularized tensor RPCA,

Cao et al. [35] designed a non-convex and non-separable model to complete background subtraction with an ADMM solver.

In this paper, a novel approach based on non-convex rank approximation minimization (NRAM) joint $l_{2,1}$ norm was proposed to overcome the deficiencies of Equation (2). Additionally, a more accurate weighted $l_1$ norm was introduced to better depict the target component. By incorporating the structured sparse item, the strong non-target component, particularly borders, could be wiped out. To solve this non-convex optimization problem, a method using alternating direction method of multipliers (ADMM) plus difference of convex (DC) programming [36] is presented, which has a lower computation complexity.

The remainder of this paper is organized as follows. In Section 2, we give an overview of related works on infrared small target detection and briefly analyze the common existing problems. The proposed NRAM model is presented in Section 3. In Section 4, extensive experiments on various scenes and sequences are conducted to illustrate the efficiency of the proposed method. Analysis and comparisons are also given. A discussion and conclusions are presented in Sections 5 and 6, respectively.

## 2. Related Works

Many algorithms have been proposed for infrared small target detection. The existing single-frame-based detection algorithms mainly fall into three main groups: methods of background-based assumption, methods of target-based assumption, and methods of target and background separating.

### 2.1. Methods of Background-Based Assumption

These methods utilize filtering to estimate the background under the assumption of background consistency. When the estimation is conducted, the target is enhanced by subtracting the background from the original image. Obviously, the selection of filters directly affects the accuracy of the detection. Tophat filter [9], Maxmean filter and Maxmedian filter [10] are three typical filters used in this detection field. By selecting different structural elements or sliding windows, targets can be caught easily in simple uniform scenes in real-time. In addition, the two-dimensional least mean square (TDLMS) filter [37] can automatically estimate the background by adaptive iteration with strong robustness compared with the above three filtering-based methods, yet does not perform in real-time. Subsequently, some improved methods, like the bilateral TDLMS (BTDLMS) filter [38], edge directional TDLMS (EDTDLMS) filter [39], and novel TDLMS filter [40] have been proposed for the sake of better performance. Unfortunately, they work well only if the background meets the simple assumption. Once the input infrared image becomes complex, the obtained residual image is frequently full of noise and background disturbance, leading to a huge challenge in identifying the true targets in post-processing. When considering complex input, they cannot handle strong target-like edges and, in fact, regard the edges as targets.

### 2.2. Methods of Target-Based Assumption

In recent years, human visual system (HVS) [11] with properties including size-adaptation, contrast mechanisms, and attention shift mechanisms have been developed to improve the performance of infrared small target detection with the assumption that the target is the most salient object. By means of filtering or transforming, the saliency map is calculated to make true targets as salient as possible, most of which are based on a single contrast mechanism such as the Laplacian of Gaussian (LoG) filter [12], and the difference of Gaussian (DoG) filter [13]. Obviously, how to define the contrast between the target and background is one of the key steps for HVS-based methods. The local contrast measure (LCM) [14] was first proposed to describe the dissimilarity between the current cell and its eight adjacent cells, which is calculated pixel by pixel and is very time consuming. For good performance in detection rate, false alarm rate and speed simultaneously, the improved LCM

(ILCM) [15], the novel LCM (NLCM) [16], and the relative LCM (RLCM) [17] were provided to calculate the saliency map. The multiscale patch-based contrast measure (MPCM) [18] focuses on the multi-directional dissimilarity between the current patch and the surrounding patches. To improve the performance of MPCM, Li et al. [41] extended it to a speed-up version, and exploited density clustering to reduce the false alarm rate. From another perspective, Li et al. [19] and Tang et al. [20] analyzed visual saliency in the frequency domain. On the whole, HSV-based methods are sensitive to edges while the small targets fail to meet the requirements of the above assumption.

### 2.3. Methods of Target and Background Separating

Apart from concentrating merely on the single background or target, some recent methods can separate the target and background simultaneously. The typical approach, IPI [21] utilizes the non-local self-correlation of infrared background and the sparsity of the target to reveal the data structure, and experimental results show an obvious superiority when compared with the traditional methods. However, it is time consuming. Dai et al. [42] pointed out that IPI would either over-shrink the small targets or leave some residuals in the target image because of the limitation of the $l_1$ norm and fixed constant weighting parameter. To address such problems, the authors proposed a weighted IPI (WIPI) model via the column-wise weights. Focusing on the defect of the nuclear norm that leads to background residuals in the target image, Dai et al. [43] analyzed the reason for the remaining residuals resulting from the mismatching of the IPI model's implicit assumption of a large amount of observations with the reality of deficient observations of strong cloud edges, and developed NIPPS exploiting partial sum minimization of singular values (PSSV) [44] to preserve the large singular values related to the observed noise data, not the whole input matrix. However, the accuracy of NIPPS depends on estimating the rank of the background matrix, whose performance degrades when facing complex backgrounds. ReWIPI [28] combines WNNM and the weighted $l_1$ norm to restrain low rank and sparsity, penalizing the larger singular values by smaller weights, so that the non-target sparse points could be efficiently suppressed.

Considering the fact that the original data were drawn from a union of low-rank subspaces, low-rank representation (LRR) [45] is used to recover background and target by a self-expressive dictionary with low-rank coefficients and the sparse noise. Based on LRR, low-rank and sparse representation (LRSR) [30] has been proposed, which adds the sparse representation of the special structure into the LRR with two additional input dictionaries. Liu et al. [46] utilized fractal background over-complete dictionary (FBOD) and generalized Gaussian target over-complete dictionary (GGTOD) together to suppress the heavy sky clutter in infrared images. Note that the projection onto the dictionary is applied to every overlapped patch, which takes lots of time, and it is difficult to generate the appropriate dictionaries. From another perspective, Wang et al. [47] presented a method named the stable multisubspace learning (SMSL) method that took into account the inner structure of actual images in heterogeneous scenes. Moreover, compared with IPI-based methods in general, multisubspace-based methods do not show superiority in terms of performance. To better recognize strong edges, Dai et al. [48] introduced structure tensor into a reweighted infrared patch-tensor model (RIPT), employing both the local and nonlocal priors. Nevertheless, it does not work well when the targets are close to or on the boundaries, leaving much noise. Worse still, this approach breaks down when the target is not sufficiently salient. Table 1 below shows a summary of the common approaches based on the separation of background and target image.

**Table 1.** Characteristics of typical methods based on target and background separation.

| Methods | Advantages | Disadvantages |
| --- | --- | --- |
| IPI | Works well with uniform scenes. | Over-shrinks the small targets, leaving residuals in the target image, time consuming. |
| NIPPS | Works well when strong edges and non-target interferences are few. | Difficult to estimate rank of data, fails to eliminate strong edges and non-target interference. |
| ReWIPI | Works well when background changes slowly. | Sensitive to rare highlight borders, performance degrading with the increasing of complexity. |
| LRR | Works well with simple scenes. | Cannot handle complex backgrounds. |
| LRSR | Works well with homogeneous backgrounds. | Difficult to choose two dictionaries simultaneously, leaving noise in target component. |
| FBOD + GGTOD | Work well with sky background clutter. | Difficult to choose two dictionaries simultaneously, cannot handle other scenes well. |
| SMSL | Works well when the target is salient and the background is clean. | Sensitive to boundaries, poor at background suppression, loses target easily. |
| RIPT | Works well when the target is salient. | Does not work well when the targets are close to or on the boundaries, leaving much noise, loses target totally when target is not sufficiently salient. |

With simple homogenous scenes, most of the listed methods can achieve good results. Table 1 shows that almost all of the approaches had a poor performance when facing real complex scenes or strong edges generated by the non-target interference sources. To improve the detection ability in complex backgrounds with non-target interference sources, we proposed a novel approach based on non-convex rank approximation minimization (NRAM) and the weighted $l_1$ norm. Figure 1 shows the recovered target image of one typical scene via IPI, NIPPS, ReWIPI, SMSL, and RIPT, respectively. Considering the common challenge that most state-of-the-art methods currently face, which is the inability to completely sweep the strong edges and interferences as observed in Figure 1, the strong edges left in the target images are of linearly structured sparsity with respect to the whole image because of the streamlined appearance of real objects and the fact that most of the buildings contain perpendicular edges [49]. The $l_{2,1}$ norm can identify the sample outliers, most of which are related to sparse structures. Hence, to better suppress strong edges, we introduce an extra regularization term on residual strong edges utilizing the $l_{2,1}$ norm.

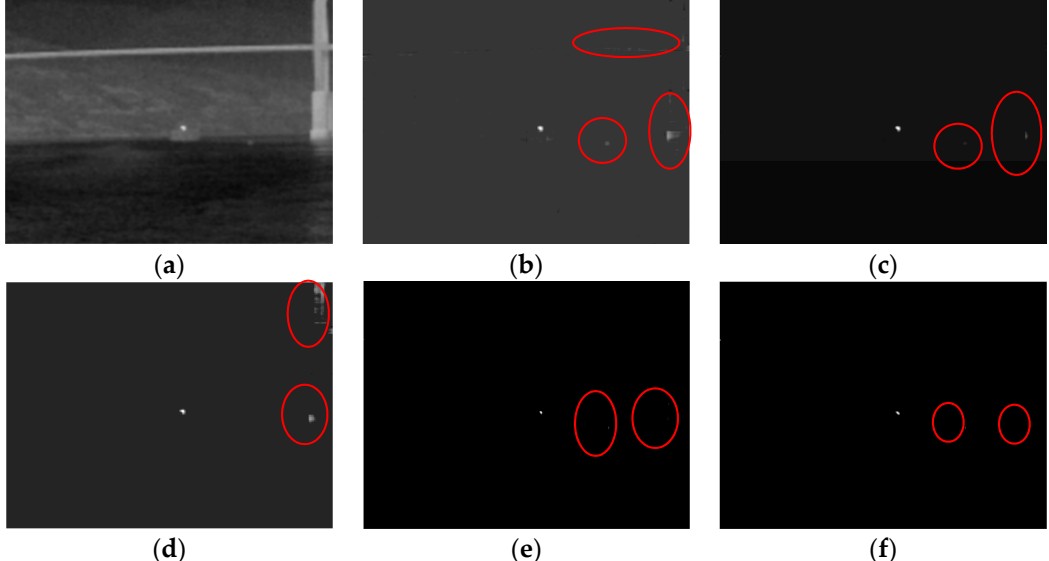

**Figure 1.** The target image recovered by five state-of-the-art methods under one typical scene. The red circle represents the near-linear background residuals left behind. (**a**) Original image; (**b**) IPI; (**c**) NIPPS; (**d**) ReWIPI; (**e**) SMSL; (**f**) RIPT.

The main contributions of this paper are as follows:

(1). We developed a novel infrared small target detection method based on non-convex rank approximation minimization and the weighted $l_1$ norm, which can approximate rank function and $l_0$ norm better, leading to a better ability to separate the target from the background than the state-of-the-art methods including IPI, NIPPS, ReWIPI, SMSL, and RIPT.

(2). Considering that most of the existing approaches suffer from residual strong edges, we introduced an additional regularization term on the remaining edges utilizing the $l_{2,1}$ norm because of the linearly structured sparsity of most interference sources.

(3). An optimization algorithm based on ADMM with DC programming was presented to solve the proposed model. Since the constraints are more powerful than the IPI-based methods, the proposed algorithm converged faster.

## 3. Proposed Method

On the whole, an infrared image with a small target can be regarded as

$$f_D = f_B + f_T + f_N \tag{3}$$

where $f_D$, $f_B$, $f_T$, $f_N$ are the original image, the background image, the target image and the noise image, respectively. As mentioned in Section 1, Gao et al. [21] presented a more general model to describe the original infrared image named IPI model based on the patch-image constructed by vectorizing the matrix within the sliding window, which is in the form of Equation (1). Since the infrared background changes slowly, this means the patches are highly correlated with both local and global patches. In other words, $B$ is a low-rank matrix. Additionally, the small target only occupies a spot of pixels, so that $T$ is a sparse matrix with respect to the whole image. To separate the background and target is to seek two solutions adapted to the original condition.

### 3.1. The Surrogate of Rank

To provide a tighter approximation than the nuclear norm does, Zhao et al. [50] proposed a novel non-convex function to directly approximate the rank named as the $\gamma$ norm. Note that the $\gamma$ norm is actually a pseudo norm. The $\gamma$ norm of matrix $B$ is

$$\|B\|_\gamma = \sum_i \frac{(1+\gamma)\sigma_i(B)}{\gamma + \sigma_i(B)}, \qquad \gamma > 0 \tag{4}$$

It can be observed that $\lim_{\gamma \to 0} \|B\|_\gamma = \mathrm{rank}(B)$ and $\lim_{\gamma \to \infty} \|B\|_\gamma = \|B\|_*$. Furthermore, for any orthonormal $U \in R^{m \times m}$ and $V \in R^{n \times n}$, $\|B\|_\gamma = \|UBV^T\|_\gamma$, namely, the $\gamma$ norm is unitarily invariant. Figure 2 indicates several surrogates of rank in the literature. Obviously, the $\gamma$ norm is almost in line with the true rank ($\gamma = 0.002$ is used here), solving the imbalanced penalization by different singular values in a traditional convex nuclear norm, while the nuclear norm deviates considerably if the singular value is far away from 1 because of the equal treatment mechanism. The log-det heuristic [51] is poor at small singular values, especially those close to 0. As can be seen, the weighted nuclear norm fits the reality better than both the nuclear norm and log-det heuristic; however, every time the weight is determined, extra singular value decomposition (SVD) occurs (see more details in Reference [24]), which increases the running time of the algorithm. To maintain both accuracy and speed, the $\gamma$ norm is the best candidate to describe the rank of a background patch-image.

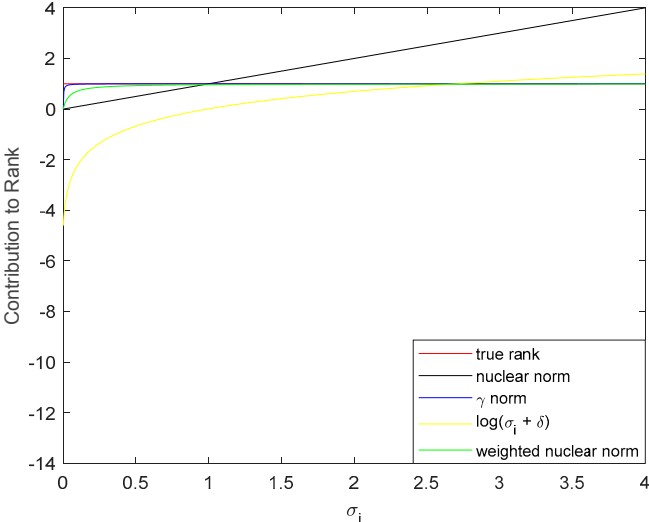

**Figure 2.** The contribution of different surrogates to the rank with respect to a varying singular value. The true rank is 1 for a nonzero $\sigma_i$.

### 3.2. The Surrogate of Sparsity

Given that dealing with the non-convex non-smooth $l_0$ norm is NP-hard, many methods [52–54] utilize the $l_1$ norm to depict the sparsity of the target patch-image. Similar to the nuclear norm, the $l_1$ norm gives every single element the same weight; this is a dilemma when there exist many strong edges and non-target sparse points. Taking into account that the brightness of most non-target sparse points is lower than that of small targets, a weighted $l_1$ norm [28] can be used to describe the target patch-image more accurately, which is given as

$$\|T\|_{W,1} = \sum_{ij} W_{ij} |T_{ij}| \tag{5}$$

$$W_{ij} = \frac{C}{|T_{ij}| + \varepsilon_T} \tag{6}$$

where $W_{ij}$ is one element of the weight matrix $W$ at position $(i, j)$; $C$ is a compromising constant and $\varepsilon_T$ is a small positive number to avoid dividing by zero.

As analyzed in Section 2, many state-of-the-art approaches fail to address the strong edges well with complex backgrounds, leaving residuals in the target images. To deal with this problem and considering the strong edges left in the target image are of linearly structured sparsity with respect to the whole image, we introduced the $l_{2,1}$ norm to efficiently remove the residuals, which is defined as

$$\|E\|_{2,1} = \sum_i \sqrt{\sum_j E_{ij}^2} \tag{7}$$

where $E$ denotes the sparse residual edges(w.r.t the whole patch-image).

Then, the proposed infrared patch-image model via non-convex rank approximation minimization (NRAM) is formulated as follows:

$$\begin{aligned} \min \quad & \|B\|_\gamma + \lambda \|T\|_{w,1} + \beta \|E\|_{2,1} \\ s.t. \quad & X = B + T + E \end{aligned} \tag{8}$$

where $\lambda$ and $\beta$ are positive trade-off coefficients.

### 3.3. Solution of the NRAM Model

An alternating direction method of multipliers (ADMM)-based optimization algorithm was devised to solve Equation (8). By introducing a Lagrange multiplier $Y$ and a penalty term, the constrained optimization problem becomes an unconstrained optimization problem and Equation (8) can be rewritten as the augmented Langrangian function:

$$L(D, B, T, E, Y, \mu) = \|B\|_\gamma + \lambda \|T\|_{w,1} + \beta \|E\|_{2,1} + \langle Y, D - B - T - E \rangle + \frac{\mu}{2} \|D - B - T - E\|_F^2 \quad (9)$$

where $\langle \bullet \rangle$ denotes the inner product of two matrices; $\|\bullet\|_F$ is the Frobenius norm. Based on ADMM, an efficient iterative approach was proposed to update $B$, $T$ and $E$, respectively. At the $(k + 1)$th step, update $B^{k+1}$, $T^{k+1}$ and $E^{k+1}$ by solving the following subproblems:

$$B^{k+1} = \underset{B}{\arg\min} \|B\|_\gamma + \frac{\mu^k}{2} \left\| D - B - T^k - E^k - \frac{Y^k}{\mu^k} \right\|_F^2 \quad (10)$$

$$T^{k+1} = \underset{T}{\arg\min} \lambda \|T\|_{W,1} + \frac{\mu^k}{2} \left\| D - B^{k+1} - T - E^k - \frac{Y^k}{\mu^k} \right\|_F^2 \quad (11)$$

$$E^{k+1} = \underset{E}{\arg\min} \beta \|E\|_{2,1} + \frac{\mu^k}{2} \left\| D - B^{k+1} - T^{k+1} - E - \frac{Y^k}{\mu^k} \right\|_F^2 \quad (12)$$

Since the $\gamma$ norm is a non-convex function and Equation (10) is a combination of non-convex and convex terms, it is appropriate to utilize difference of convex (DC) programming to solve Equation (10) (for details please see [50]). The DC algorithm regards the original non-convex problem as the difference of two convex functions and optimizes it iteratively by linearizing the concave term at each iteration. At the $(t + 1)$th inner iteration

$$\sigma^{t+1} = \underset{\sigma \geq 0}{\arg\min} \langle \omega_t, \sigma \rangle + \frac{\mu^k}{2} \|\sigma - \sigma_A\|_2^2 \quad (13)$$

which admits a closed-form solution

$$\sigma^{t+1} = \max(\sigma_A - \frac{\omega_t}{\mu^k}, 0) \quad (14)$$

where $A = D - T^k - E^k - \frac{Y^k}{\mu^k}$, $\omega_t = \frac{(1+\gamma)\gamma}{(\gamma + \sigma^t)^2}$ and $U\mathrm{diag}\{\sigma_A\}V^\mathrm{T}$ is the SVD of $A$. After a few iterations (actually within two iterations), it converges to a local optimal point $\sigma^*$. Then, $B^{k+1} = U\mathrm{diag}\{\sigma^*\}V^\mathrm{T}$.

According to [55] and [28], Equations (11) and (12) can be solved as

$$T^{k+1} = S_{\lambda W/\mu^k} \left( D - B^{k+1} - E^k - \frac{Y^k}{\mu^k} \right) \quad (15)$$

$$[E^{k+1}]_{:,i} = \begin{cases} \frac{\|Q_{:,i}\|_2 - \frac{\beta}{\mu^k}}{\|Q_{:,i}\|_2} Q_{:,i} & \text{if } \|Q_{:,i}\|_2 > \frac{\beta}{\mu^k} \\ 0 & \text{otherwise} \end{cases} \quad (16)$$

where $Q = D - B^{k+1} - T^{k+1} - \frac{Y^k}{\mu^k}$ and $[E^{k+1}]_{:,i}$ is the $i$-th column of $E^{k+1}$.

$Y$ and $\mu$ updates in the standard way:

$$Y^{k+1} = Y^k + \mu^k(D - B^{k+1} - T^{k+1} - E^{k+1}) \quad (17)$$

$$\mu^{k+1} = \rho\mu^k \quad (18)$$

where $\rho > 1$. Finally, the whole process is described in Algorithms 1 and 2.

---

**Algorithm 1:** ALM/DC solver to the NRAM model

---

**Input:** Original patch-image $D$, $\lambda$, $\beta$, $\mu^0$, $\gamma$;
**Output:** $B = B^k + E^k$, $T = T^k$;
**Initialize:** $B^0 = T^0 = E^0 = Y^0 = 0$, $\rho = 1.1$, $\varepsilon = 10^{-7}$, $k = 0$, $W^0 = 1 * 1^T$, $\Lambda^0 = 0 \in R^{\min(m,n) \times 1}$;
**While** not converged **do**
1: Fix the others and update $B$ by DC programming;
2: Fix the others and update $T$ by

$$T^{k+1} = S_{\lambda W/\mu^k}(D - B^{k+1} - E^k - \frac{Y^k}{\mu^k});$$

3: Fix the others and update $E$ by

$$[E^{k+1}]_{:,i} = \begin{cases} \dfrac{\|Q_{:,i}\|_2 - \dfrac{\beta}{\mu^k}}{\|Q_{:,i}\|_2} Q_{:,i} & \text{if } \|Q_{:,i}\|_2 > \dfrac{\beta}{\mu^k} \\ 0 & \text{otherwise} \end{cases};$$

4: Fix the others and update $Y$ by
　　$Y^{k+1} = Y^k + \mu^k(D - B^{k+1} - T^{k+1} - E^{k+1});$
5: Update $W$ by

$$W_{ij} = \frac{C}{\left|T_{ij}\right| + \varepsilon_T};$$

6: Update $\mu$ by
　　$\mu^{k+1} = \rho\mu^k;$
7: Check the convergence conditions

$$\frac{\|D - B^{k+1} - T^{k+1} - E^{k+1}\|_F}{\|D\|_F} < \varepsilon;$$

8: Update $k$;
　　$k = k + 1;$
**End while**

---

**Algorithm 2:** DC programming

---

**Input:** $B^k$, $\gamma$, $A = D - T^k - E^k - \dfrac{Y^k}{\mu^k}$, $\Lambda^0$;

**Output:** $B^{k+1}$, $\Lambda^{t+1}$;
**Initialize:** $t = 0$, $[U, S, V] = \text{svd}(A)$;
**While** not converged **do**
1: Calculate
　　$\omega_t = \dfrac{(1 + \gamma)\gamma}{(\gamma + \Lambda^t)^2}$, $\Lambda^{t+1} = \max(S - \dfrac{\omega_t}{\mu^k}, 0);$
2: Check the convergence conditions
　　$\|\Lambda^{t+1} - \Lambda^t\|_2 < \varepsilon;$
**End while**
3: $B^{k+1} = U\Lambda^{t+1}V;$

---

### 3.4. The Whole Process of the Proposed Method

Figure 3 shows the whole process of the proposed method in this paper, which can be summarized as follows:

(1). Patch-image construction. By sliding a window of size $k \times k$ from top to bottom and from left to right to transform the original infrared image $I \in R^{m \times n}$ into a patch-image $D \in R^{k^2 \times t}$, where $t$ is the number of window slips, the matrix inside each matrix was vectorized as a column of the constructed patch-image $D$.

(2).　Target-background separation. The input patch-image *D* was decomposed into a low-rank matrix *B*, a sparse matrix *T* and a structural noise (strong edges) matrix *E*. Since the structural noise belongs to the background, we summed *B* and *B* as the final recovered background patch-image *B*.

(3).　Image reconstruction and target detection. Reconstruction is the inverse process of construction. For the position overlapped by several patches, we utilized the one-dimensional median filter to determine the value. Small targets were detected by adaptive threshold segmentation; the selection of the threshold was based on [21].

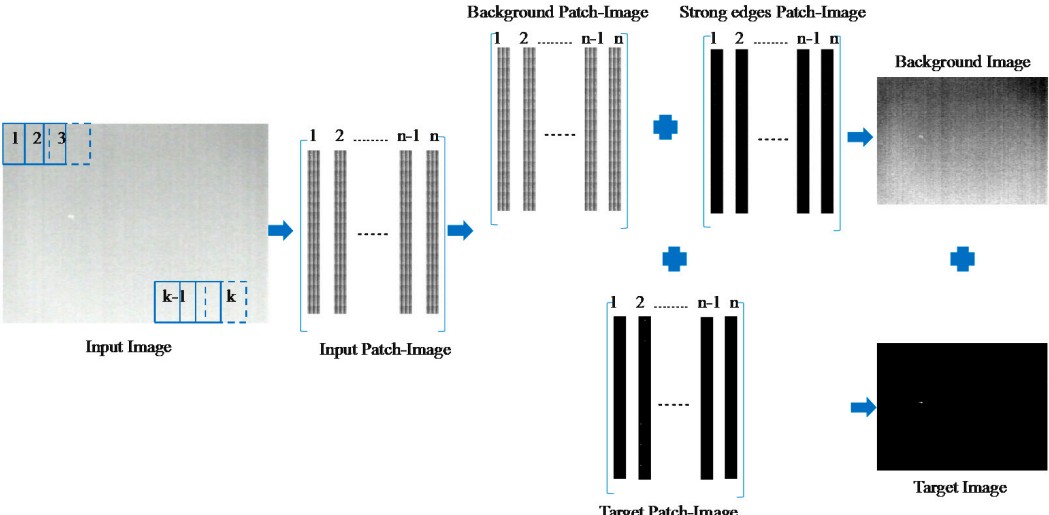

**Figure 3.** The overall procedure of the roposed model in this paper.

## 4. Experiment and Analysis

### 4.1. Experimental Preparation

In this section, eight single practical scenes and six real sequences (the scene of each sequence was contained in the eight tested scenes) were used to test the validity of the proposed method. The eight scenes are shown in Figure 4. As most of the existing algorithms have a good effect on an infrared image with a salient target and clean background, the six sequences were not easy to deal with. Sequences 1–4 are characterized by complex backgrounds with salient strong edges and interference sources. Although both of the backgrounds were homogeneous and uniform, Sequences 5 and 6 are actually difficult because of the dim targets, which also shows the adaptability of the proposed method to different scenes. A detailed description is given in Table 2. Meanwhile, eight other state-of-the-art methods including Tophat transformation [9], local contrast measure (LCM) [14], multiscale patch-based contrast measure (MPCM) [18], infrared patch-image model (IPI) [21], non-negative infrared patch-image model based on partial sum minimization of singular values (NIPPS) [43], reweighted IPI (ReWIPI) [28], stable multisubspace learning (SMSL) [47], reweighted infrared patch-tensor model (RIPT) [48] were employed as the baselines for the sake of comparison. All parameters of the seven approaches given by the authors were fine tuned for each specific sequence due to the lack of robustness of some of the approaches in addressing complex backgrounds. The final parameter settings are presented in Table 3. In addition, all experiments were performed with Matlab R2018a in Windows 7 based on an Intel Celeron 2.90 GHz GPU with 4 G RAM (The code is available at https://github.com/Lanneeee/NRAM).

**Table 2.** Detailed description of the eight real scenes.

| Sequence | Frame Number | Size | Background Description | Target Description |
|---|---|---|---|---|
| Scene 1 (Sequence 1) | 50 | $208 \times 152$ | Homogenous with a highlight punctate disturbance | Small, very dim with low contrast |
| Scene 2 (Sequence 2) | 67 | $320 \times 240$ | Very bright, heavy noise | Moves fast with changing shape, brightness |
| Scene 3 (Sequence 3) | 52 | $128 \times 128$ | Sky scene with banded cloud | Tiny |
| Scene 4 (Sequence 4) | 30 | $256 \times 200$ | Heavy banded cloud and floccus | Small, size varies a lot |
| Scene 5 (Sequence 5) | 200 | $252 \times 220$ | Complex background with trees and buildings | Quite small size, changing slightly in the sequence |
| Scene 6 (Sequence 6) | 185 | $252 \times 213$ | Swinging plants that obscure the target frequently | Small, keeps moving in the sequence and changing brightness |
| Scene 7 | — | $320 \times 240$ | Sea and mountain scene with obvious artificial structure | Ship target, bright, Gaussian shape |
| Scene 8 | — | $128 \times 128$ | Low contrast with a lot of noise | Tiny and not so salient |

**Table 3.** Detailed parameter settings of the nine tested methods.

| Method | Parameters |
|---|---|
| Tophat | Structure shape: disk, structure size: $3 \times 3$ |
| LCM | Cell size: $3 \times 3$ |
| MPCM | $N = 3, 5, 7, 9$, mean filter size: $3 \times 3$ |
| IPI | Patch size: $50 \times 50$, sliding step: 10, $\lambda = 1/\sqrt{\min(m,n)}$, $\varepsilon = 10^{-7}$ |
| NIPPS | Patch size: $50 \times 50$, sliding step: 10, $\lambda = 1/\sqrt{\min(m,n)}$ for sequence 3, $\lambda = 2/\sqrt{\min(m,n)}$; for the rest, $\varepsilon = 10^{-7}$ |
| ReWIPI | Patch size: $50 \times 50$, sliding step: 10, $\lambda = 1/\sqrt{\min(m,n)}$ for sequence 3, $\lambda = 2/\sqrt{\min(m,n)}$; for the rest, $\varepsilon = 10^{-7}$, $\varepsilon_B = \varepsilon_T = 0.08$ |
| SMSL | Patch size: $30 \times 30$, sliding step: 10, $\lambda = 3/\sqrt{\min(m,n)}$, $\mu^0 = 0.5 * s_4$, $\overline{\mu} = 0.05 * s_5$, $\varepsilon = 10^{-7}$ |
| RIPT | Patch size: $30 \times 30$, sliding step: 10, $\lambda = L/\sqrt{\min(m,n)}$, $L = 0.7$, $h = 1$, $\varepsilon = 10^{-7}$ |
| NRAM | Patch size: $50 \times 50$, sliding step: 10, $\lambda = 1/\sqrt{\min(m,n)}$, $\mu^0 = 3\sqrt{\min(m,n)}$, $\gamma = 0.002$, $C = \sqrt{\min(m,n)}/2.5$, $\varepsilon = 10^{-7}$ |

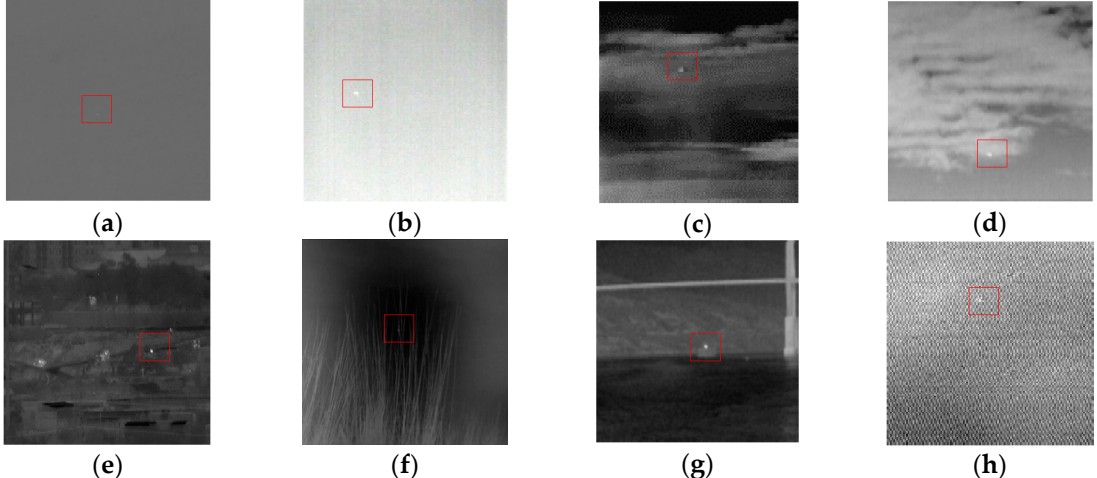

**Figure 4.** The eight real scenes used in the experiments. For the sake of visualization, all the images are changed to the same size. (**a**) Scene 1 (Sequence 1); (**b**) Scene 2 (Sequence 2); (**c**) Scene 3 (Sequence 3); (**d**) Scene 4 (Sequence 4); (**e**) Scene 5 (Sequence 5); (**f**) Scene 6 (Sequence 6); (**g**) Scene 7; (**h**) Scene 8.

### 4.2. Evaluation Metrics

For a comprehensive comparison, the whole evaluation process was based on qualitative and quantitative evaluations. All the separated target images obtained by the different methods were evaluated via several commonly used metrics to show the ability of background suppression and target enhancement, namely the signal-to-clutter ratio gain (SCRG), the background suppression factor (BSF) and the receiver operating characteristic (ROC) curve.

SCRG is the most widely used criterion whose definition is

$$SCRG = \frac{SCR_{out}}{SCR_{in}} \tag{19}$$

where SCR is a measurement of detection difficulty and target saliency; the subscripts out and in denote the original image and the separated target image, respectively. SCR is defined as

$$SCR = \frac{|\mu_t - \mu_b|}{\sigma_b} \tag{20}$$

where $\mu_t$ and $\mu_b$ are the average pixel value of the target region and surrounding local neighborhood region and $\sigma_b$ is the standard deviation of the surrounding local neighborhood region. The local region used in the experiment is illustrated in Figure 5. The size of the target is $a \times b$, the local region size is $(a + 2d) \times (b + 2d)$, we set $d = 40$ in this paper.

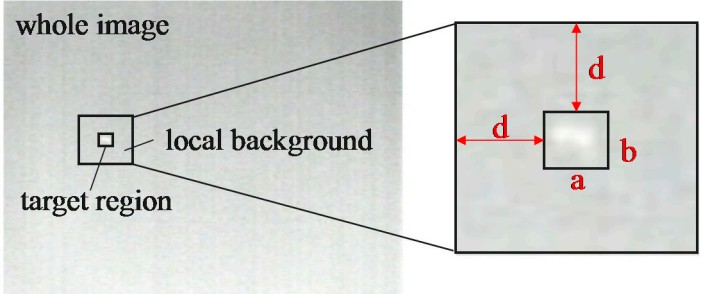

**Figure 5.** Local region of a small target in an infrared image.

Another widely used metric is BSF, showing the performance of background suppression, which is defined as

$$\text{BSF} = \frac{\sigma_{in}}{\sigma_{out}} \tag{21}$$

where $\sigma_{in}$ and $\sigma_{out}$ represent the corresponding standard variances of the background neighborhood before and after processing. Higher SCRG and BSF values mean the better performance of an algorithm.

Besides these two metrics, the detection probability $P_d$ and false-alarm rate $F_a$ are a pair of indicators for comprehensive detection ability. $P_d$ and $F_a$ are defined as follows:

$$P_d = \frac{\text{number of true detections}}{\text{number of actual targets}} \tag{22}$$

$$F_a = \frac{\text{number of false detections}}{\text{number of images}} \tag{23}$$

The ROC curve shows the trade-off between the true detections and false detections.

### 4.3. Parameter Analysis

There are several vital parameters like the penalty factor $\mu$, the compromising constant $C$ that determines the weight coefficient matrix and the norm factor $\gamma$ that affects the low-rank approximation. Hence, to obtain a better performance with real datasets, it is necessary to choose proper parameters. The ROC curves corresponding to different parameters are shown in Figure 6 for Sequences 1–5. It should be noted that the performances obtained by tuning one of the parameters with the others fixed may not be globally optimal.

#### 4.3.1. Penalty Factor $\mu$

$\mu$ has a direct impact on the whole process, especially the soft-thresholding operator which determines the target component, thus one has to choose $\mu$ properly in order to ensure both optimality and fast convergence rate. A smaller $\mu$ would improve the ability of preserving details in the background image; however, the details of the target might also be recovered, leading to over-shrinking of the small target or no target in the target image. On the other hand, a larger $\mu$ can protect the target but would also leave more noise in the target image. In fact, the small singular value corresponds to the noise in the image, as illustrated in Figure 2; the $\gamma$ norm fit the true rank well in most cases but when it came to the singular value that approached zero, the weight given to it dropped sharply until reaching zero, leading to an over-smooth background and a noisy target image. To address this problem, a larger $\mu$ is needed. Furthermore, a larger $\mu$ conducts a faster convergence. In order to change $\mu$ adaptively, we set $\mu = c\sqrt{\min(m, n)}$, where $m$ and $n$ are the length and width of patch image and $c$ is a positive constant. To figure out the influence of the penalty factor on Sequences 1–5, instead of changing $\mu$ directly, we varied $c$ from 0.5 to 5. For all 5 sequences, $c = 2$ or 3 achieved the best performance, while $c = 0.5$ or 5 performed the worst. This is because a smaller $\mu$ would lose the target and a too large $\mu$ would regard the salient non-target noise as the "true" target, also resulting in incorrect recovery in most cases, especially when the target was not so prominent. In fact, $c = 2$ shrank the target a little. We chose $c = 3$ as the best value.

#### 4.3.2. Norm Factor $\gamma$

$\gamma$ is an important factor determining the ability of background recovery. With smaller $\gamma$, the rank surrogate $\gamma$ norm closely matches the true rank. However, if $\gamma$ is too small, the target would also be treated as a low-rank component. Apparently, $\gamma$ cannot be too large as that leads to deviation from the real rank, just as the nuclear-norm does. Thus, it is important to find an appropriate value for $\gamma$ to keep the balance between detection probability and the false-alarm ratio. We show the effects of $\gamma$ in the third and fourth rows of Figure 6. We varied the value of $\gamma$ from 0.001 to 0.005 with an interval of

0.001. It can be observed that even if $\gamma$ changes slightly, this change has a great impact on the results. $\gamma = 0.002$ remained at a stable level while the rest had some fluctuations. The larger the $\gamma$, the worse the performance, thus the best choice for $\gamma$ was 0.002.

### 4.3.3. Compromising Constant *C*

$C$ plays an important role in controlling the adaptive weight value, which has to be chosen carefully. With reference to [24] with slight modification, we set $C$ as $\alpha\sqrt{\min(m,n)}$ in our experiment for the sake of self-adaption. The performance of different $\alpha$ is shown in the last two rows of Figure 6, from which we can conclude that the algorithm was quite insensitive to the variation of $C$. Different $C$ all performed well on Sequence 5 resulting from the target being easy to detect while the non-target noise was hard to suppress. Synthesizing the results of all sequences, we set $\sqrt{\min(m,n)}/2.5$ as the optimum value.

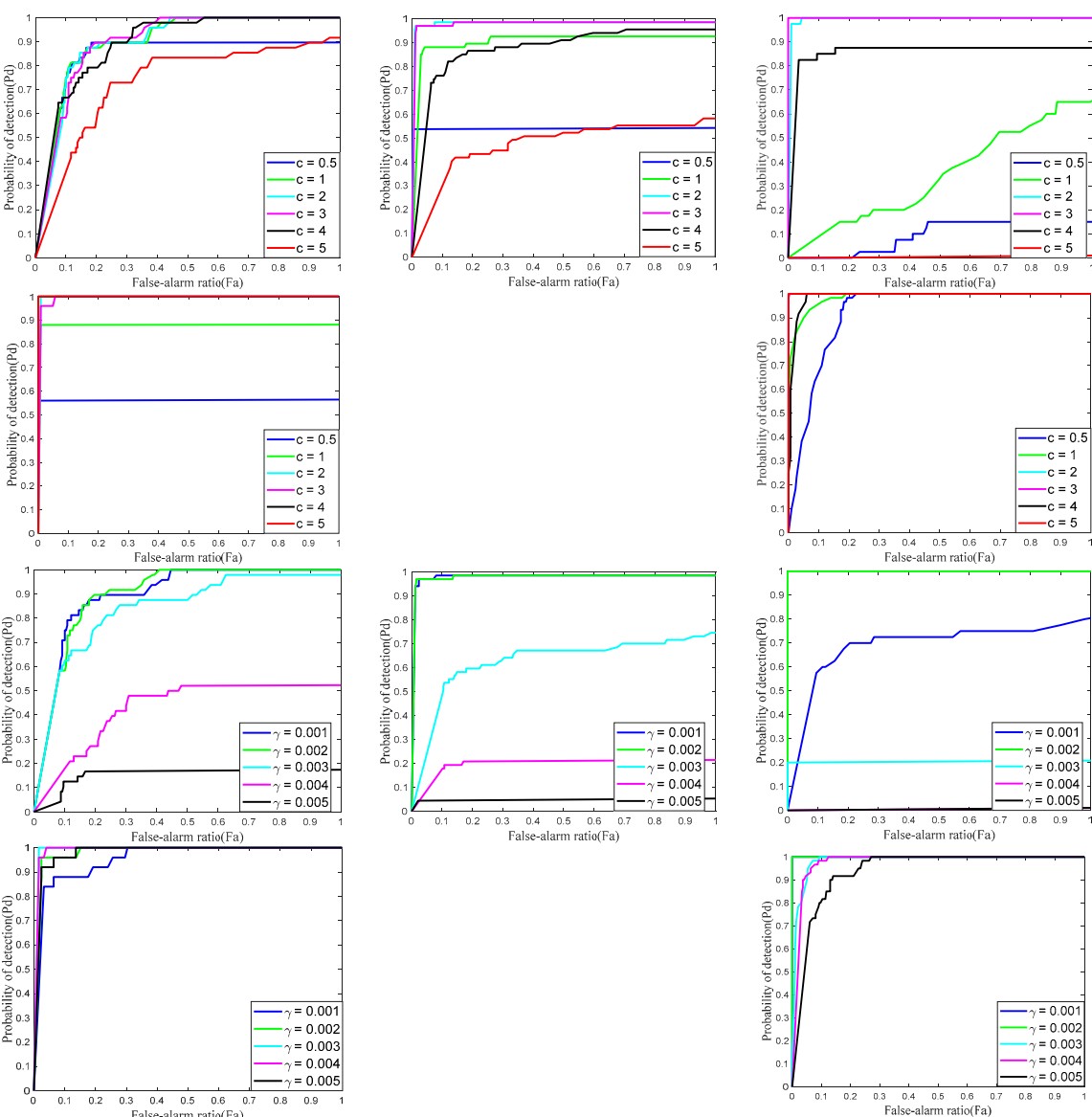

**Figure 6.** *Cont.*

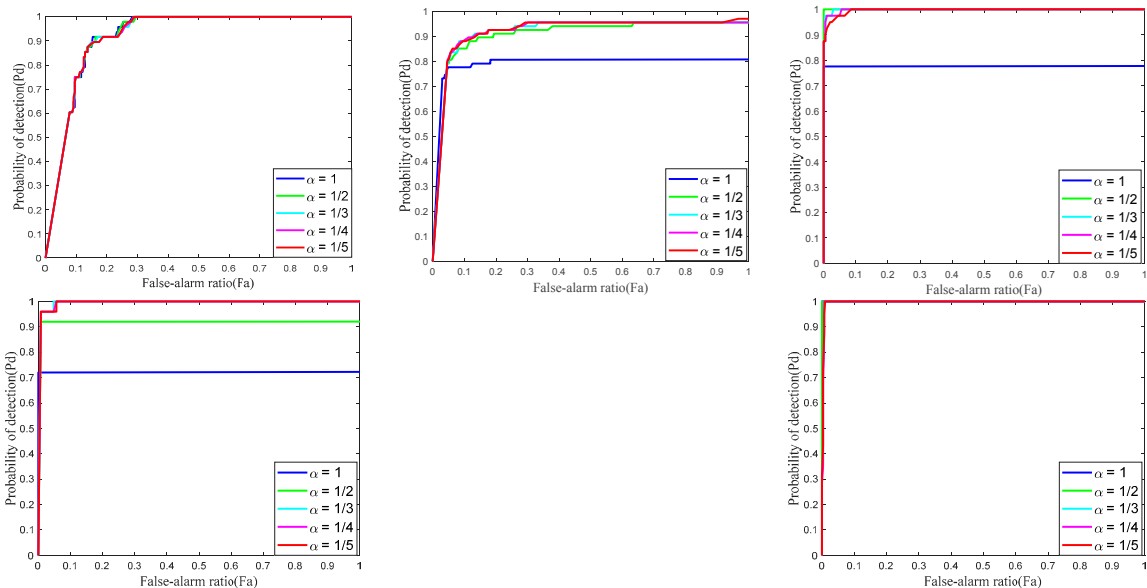

**Figure 6.** Detection performances under different parameters. Rows 1–2: ROC curves with respect to different penalty factors, Rows 3–4: ROC curves with respect to different norm factors, Rows 4–5: ROC curves with respect to different compromising constants.

### 4.4. Qualitative Evaluation under Different Scenes

In this section, the proposed method was compared with eight state-of-the-art methods in background suppression performance. Figures 7–9 are the results obtained by all the tested methods under different scenes. Note that the target has been marked with a red box.

The Tophat operator did enhance the target; however, it was extremely sensitive to edges and noise leading to a high false-alarm ratio, especially in the case of a complex background such as in Figures 8A, 9B and 10B. The main reason comes down to the fixed structure element and the hypothesis of smooth background. There is no doubt that the performance of the LCM was the worst among all results produced by the tested methods. This is because the contrast mechanism was not so applicable with a complex background. Moreover, when the local region of the target had low contrast, some other strong edges or brighter non-target components were highlighted instead. The worst was in Figure 10B because the contrast was almost uniform everywhere. Unlike LCM, another HSV-based approach, MPCM, defines the local contrast measure based on the difference between the reference patch and the background patches and not on the ratio of the maximum grayscale of the central cell to the average value of the surrounding cells. MPCM worked much better when facing different backgrounds, particularly in clean and uniform scenes such as in Figure 7 and showed more robustness when compared with LCM. Nevertheless, from Figures 8–10, we can see that it still suffered from salient non-target objects caused by the inaccuracy of the local dissimilarity measure. In other words, the simple local dissimilarity measure or assumption based on background consistency cannot deal with various scenes.

The other five baselines aim at separating the background and target by solving optimization problems, which have relative superiority over conventional approaches. From the obtained target images, it was clear that IPI could recover the target well while also preserving the background residuals because of the deficiencies of both the nuclear norm and $l_1$ norm, as analyzed in Section 1. This phenomenon was most conspicuous when the target was very dim and the background edge was abundant such as in Figure 8A. NIPPS was better than IPI and could suppress most of the strong clutter and keep the shape of the target by minimizing the partial sum of singular values. Still, as observed in Figure 8, complex scenes including highlight interference sources and intensive noise is a challenge for NIPPS. By observing Figures 8–10, it can be said with certainty that the NIPPS model does not satisfy scenario robustness. Furthermore, the rank estimation task was harder to address

when salient clusters and non-target components exist, as shown in Figures 9A and 10B. Based on the weighted nuclear norm minimization, ReWIPI applies weighted technology to restore background and target simultaneously. However, this method showed little improving effect given that, considering the backgrounds in Figure 9, the residual problem of sparse edges remained unsolved. Moreover, the convergence rate of ReWIPI was slow, resulting in high computation complexity, which will be further discussed later.

SMSL and RIPT had an unstable performance as concluded from their produced target image. From Figures 7, 8A and 9A, both approaches lost the target totally. For SMSL, when the target was so dim that it was difficult to distinguish the target from the background clearly, the learning mechanism would assign the target to a subspace of the background, thus the target would disappear. In addition, in Figure 8B, spot-like components lying on the boundaries would also bring about false detection because of the inaccuracy of the subspace construction. With the purpose of wiping out disturbance caused by complex backgrounds, RIPT utilizes the local structure tensor to suppress the salient background edges, which need an obvious structure. RIPT indeed achieved a good performance with some heterogeneous scenes such as in Figures 8B and 10A, with the help of the structure tensor. Unfortunately, like in Figure 7, a dim target overwhelmed by background clutter did not meet the requirement and fuzzy edges even caused strange detection results. Merely using the single prior of background and not incorporating it with the target prior also explained the bad results.

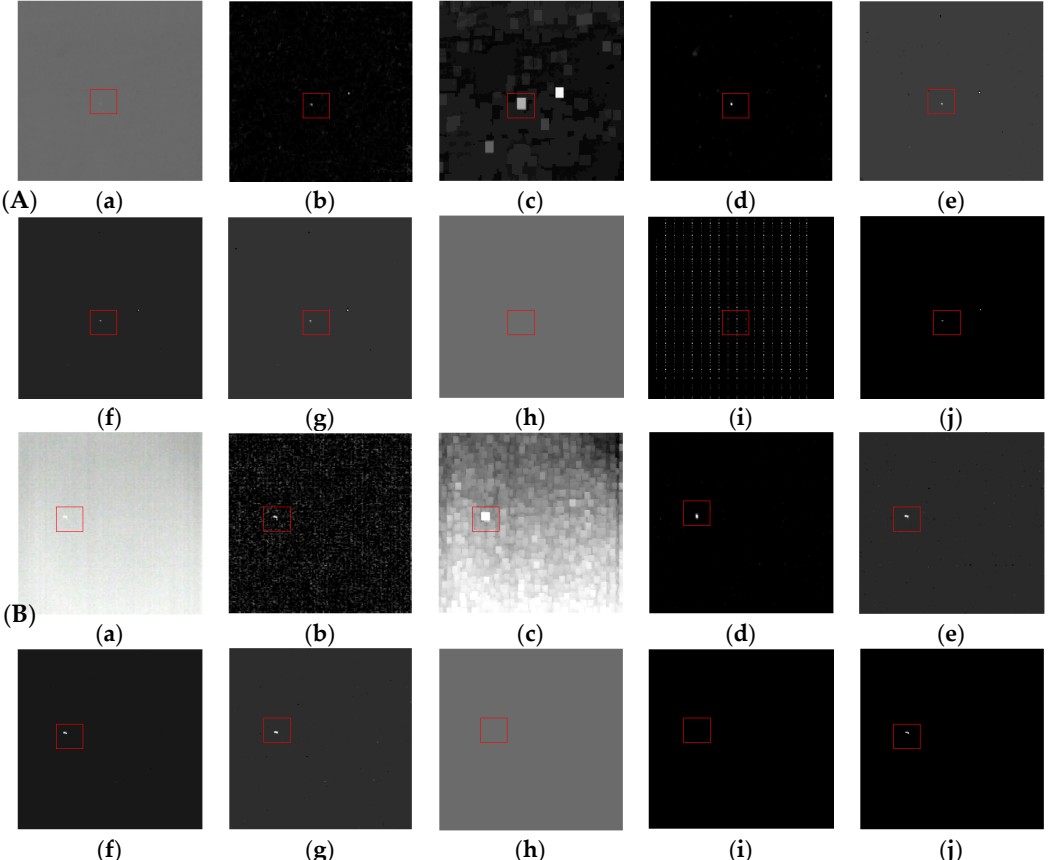

**Figure 7.** Results of the different approaches to Scenes 1 and 2. (**A**) Scene 1 (Sequence 1); (**B**) Scene 2 (Sequence 2); (**a**) Original image; (**b**) Tophat; (**c**) LCM; (**d**) MPCM; (**e**) IPI; (**f**) NIPPS; (**g**) ReWIPI; (**h**) SMSL; (**i**) RIPT; (**j**) NRAM.

Compared with all of the above baselines, it can be concluded that the proposed NRAM model provided satisfactory performances in various scenes not only in keeping the shape of the target but also in wiping out rare clutter, especially in Figures 8–10 whose targets were dim and the disturbances

were strong. In Figure 10B, when the target was seriously disturbed by noise, all baselines failed to detect the target and only the proposed NRAM worked relatively well even though the given target had been shrunk because of the overwhelming clutter. This demonstrates the strong scenario robustness that is lacking in the other state-of-the-art methods.

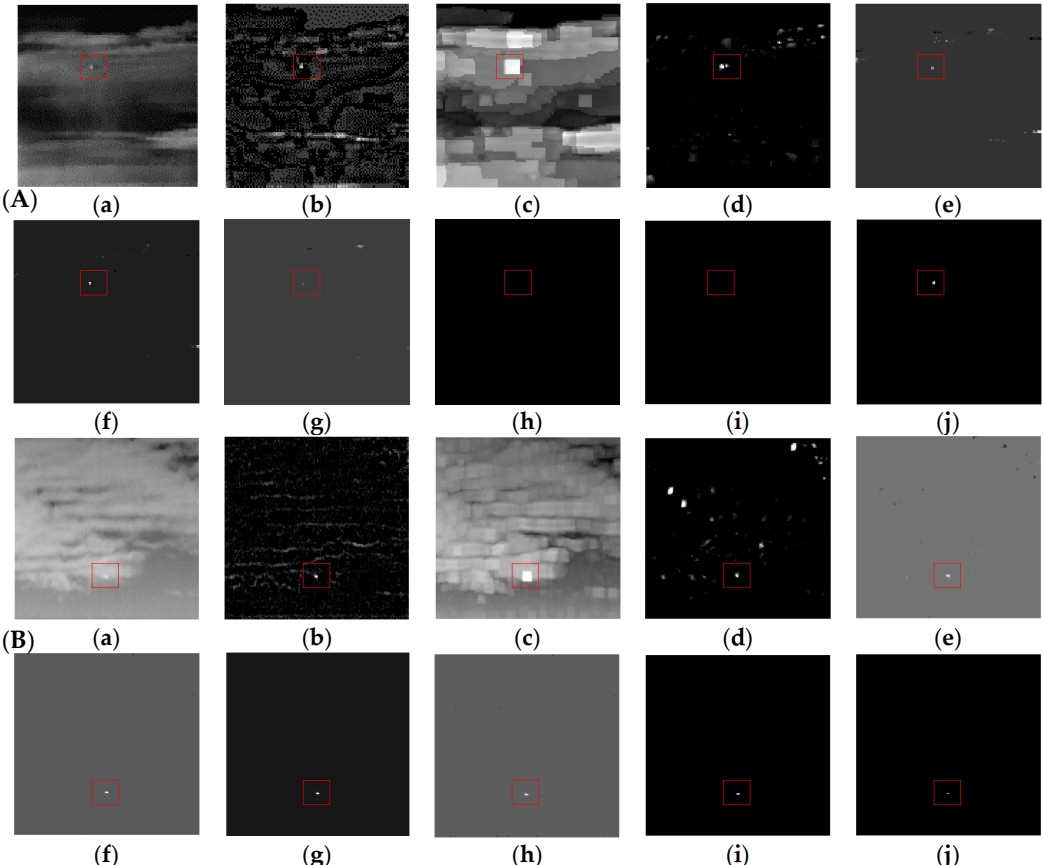

**Figure 8.** Results of different approaches to Scenes 3 and 4. (**A**) Scene 3 (Sequence 3); (**B**) Scene 4 (Sequence 4); (**a**) Original image; (**b**) Tophat; (**c**) LCM; (**d**) MPCM; (**e**) IPI; (**f**) NIPPS; (**g**) ReWIPI; (**h**) SMSL; (**i**) RIPT; (**j**) NRAM.

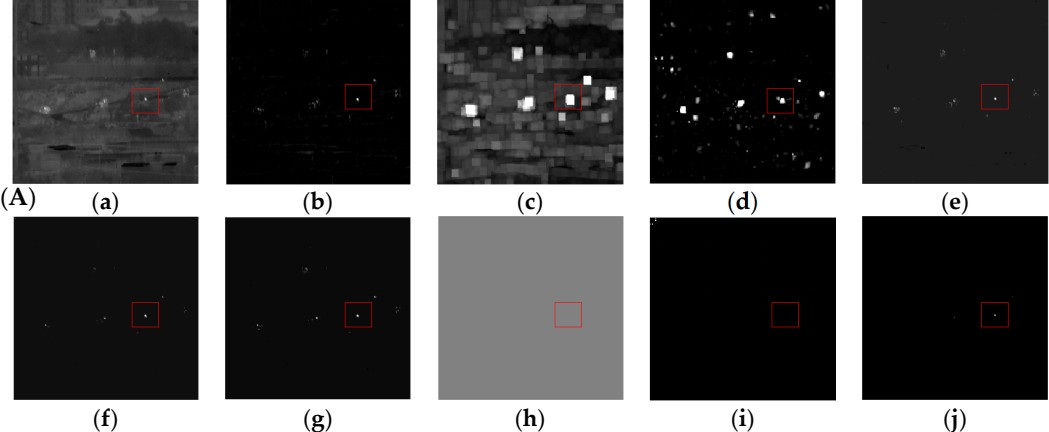

**Figure 9.** *Cont.*

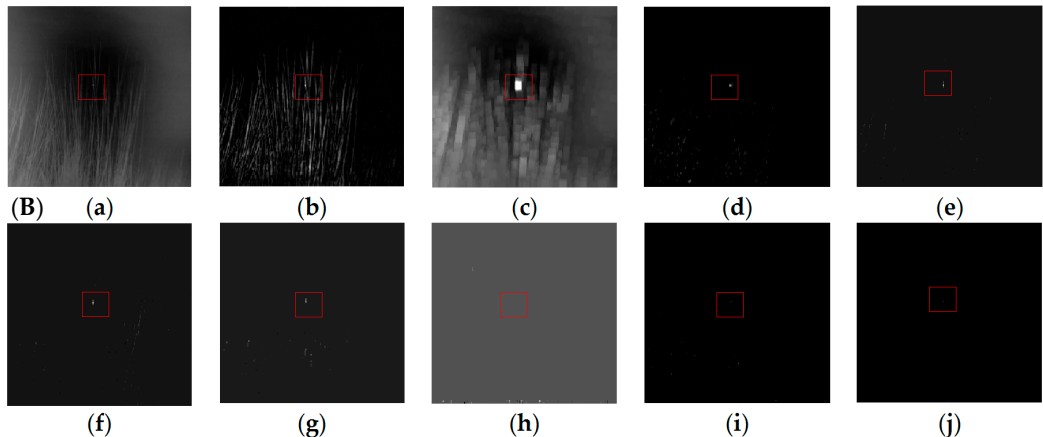

**Figure 9.** Results of different approaches to Scene 5 and 6. (**A**) Scene 5 (Sequence 5); (**B**) Scene 6 (Sequence 6); (**a**) Original image; (**b**) Tophat; (**c**) LCM; (**d**) MPCM; (**e**) IPI; (**f**) NIPPS; (**g**) ReWIPI; (**h**) SMSL; (**i**) RIPT; (**j**) NRAM.

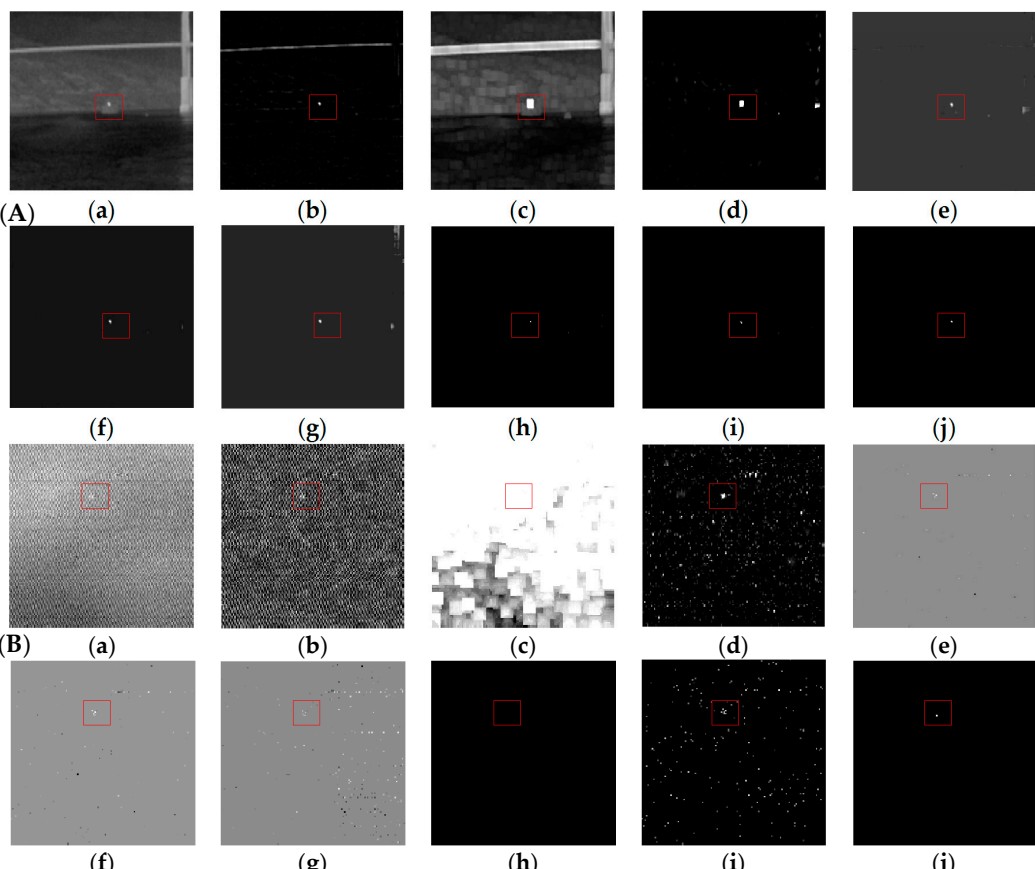

**Figure 10.** Results of different approaches to Scenes 7 and 8. (**A**) Scene 7; (**B**) Scene 8; (**a**) Original image; (**b**) Tophat; (**c**) LCM; (**d**) MPCM; (**e**) IPI; (**f**) NIPPS; (**g**) ReWIPI; (**h**) SMSL; (**i**) RIPT; (**j**) NRAM.

To visualize the complexity of the scenes and to compare the effectiveness of the other seven methods more intuitively, 3D gray maps of several representative tested images and obtained images (Figures 8A,B and 10B) are displayed in Figure 11. It was obvious that the three scenes were all complex. From Figure 11, we can see that background components usually exist with a brightness close to or above the targets, which makes it more difficult to detect infrared dim targets accurately. MPCM enhanced the targets as well as the clutter, leaving a locating challenge. IPI, NIPPS and ReWIPI

performed relatively well in most cases but there were two drawbacks: first, some strong clutter was not removed; and second, the degree of background suppression was not high (in other words, in the 3D gray maps, the minimum gray value did not reach 0). In contrast, the proposed NRAM model almost completely suppressed the background and clutter with fewer false alarms, showing strong robustness. Moreover, in the third scene, all but the proposed method were not successful. Hence, from a qualitative perspective, the NRAM model has huge advantages.

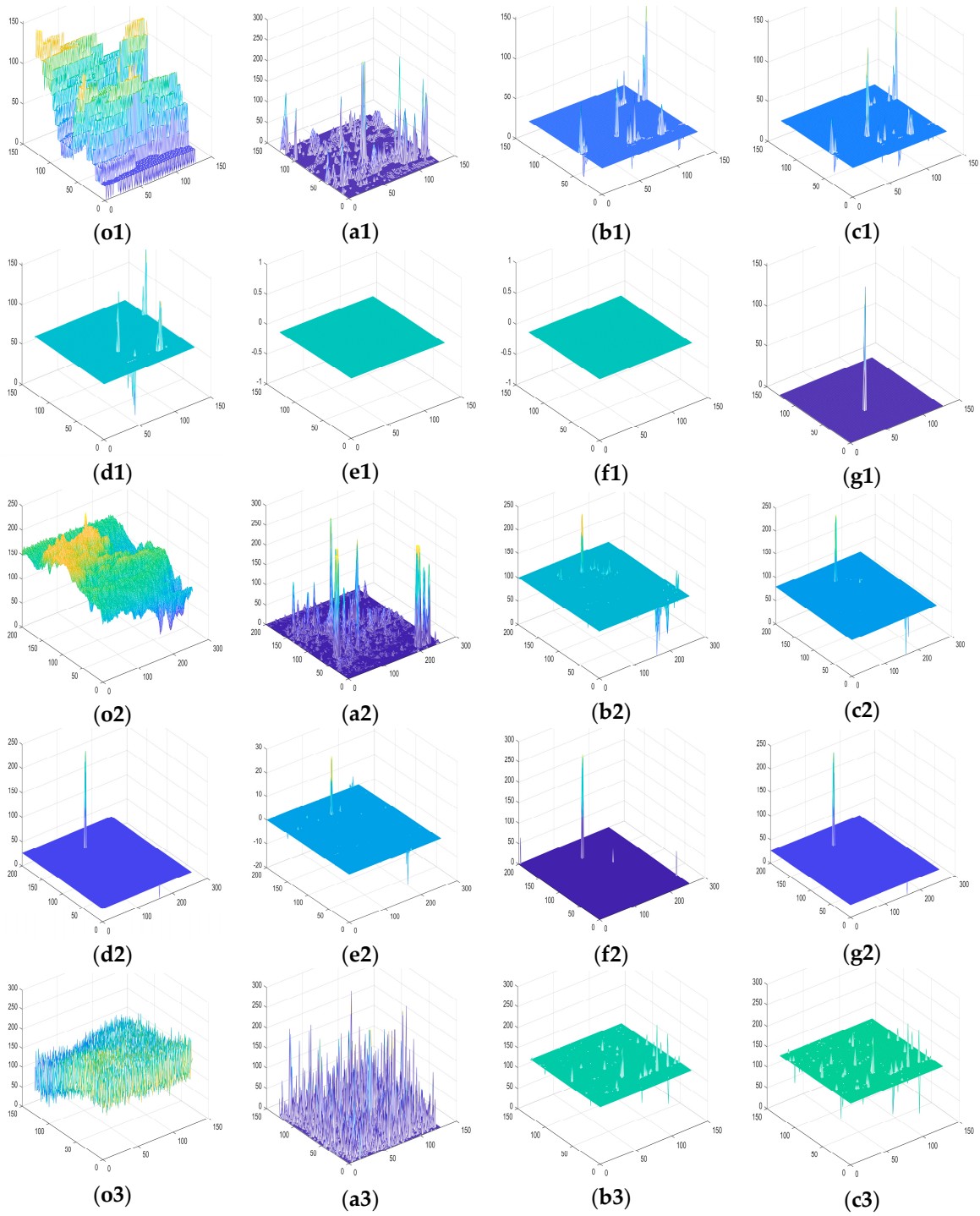

**Figure 11.** *Cont*.

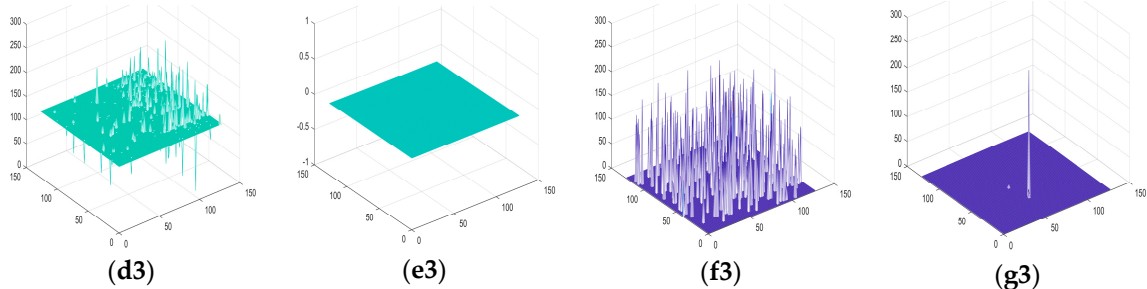

**Figure 11.** 3D gray maps of three representative scenes and corresponding results. (**a**) Original image; (**b**) MPCM; (**c**) IPI; (**d**) NIPPS; (**e**) SMSL; (**f**) RIPT; (**g**) NRAM.

*4.5. Quantitative Evaluation of Sequences*

Aside from the qualitative evaluation based on different scenes, to evaluate the ability of background suppression and target enhancement in a quantitative way, evaluation indices including signal-to-clutter ratio gain (SCRG) and background suppression factor (BSF) were used to assess the performance. The experimental results of six actual sequences (Figures 7–9) are shown in Table 4. Note that because SMSL and RIPT failed in most cases, the corresponding quantitative values were not calculated. It can be seen that the proposed method almost always achieved the highest values, showing great superiority in background suppression. In general, as before, the last six methods were better than the former simple assumption-based methods and there was no surprise that the LCM obtained the lowest scores.

**Table 4.** SCRG and BSF values of the nine methods.

| Method | Seq 1 SCRG BSF | Seq 2 SCRG BSF | Seq 3 SCRG BSF | Seq 4 SCRG BSF | Seq 5 SCRG BSF | Seq 6 SCRG BSF |
|---|---|---|---|---|---|---|
| Tophat | 4.92 5.55 | 2.09 10.66 | 1.54 7.55 | 2.29 8.59 | 3.78 3.00 | 0.53 0.47 |
| LCM | 0.49 0.69 | 0.75 0.93 | 2.17 1.18 | 0.25 0.69 | 0.40 0.96 | 0.91 0.95 |
| MPCM | 4.56 5.49 | 2.05 8.20 | 1.22 6.85 | 0.49 1.96 | 2.59 2.50 | 2.44 3.76 |
| IPI | 7.36 2.25 | 7.64 2.10 | 2.59 4.46 | 4.24 5.08 | 2.10 4.29 | 3.25 3.08 |
| NIPPS | 35.85 14.32 | 11.48 4.42 | 2.89 7.30 | 4.03 5.61 | 3.94 11.84 | 6.29 3.95 |
| ReWIPI | 10.71 6.72 | 11.01 2.93 | 3.26 8.36 | 3.40 3.18 | 2.03 3.93 | 2.52 2.58 |
| SMSL | — | 1.86 10.34 | 2.80 2.99 | — | — | — |
| RIPT | — | 4.39 23.39 | 2.12 11.5 | — | — | — |
| NRAM | 87.34 60.30 | 11.02 50.01 | 2.72 21.29 | 8.81 25.70 | 7.75 32.58 | 7.87 59.21 |

NOTES: Red represents the highest value and blue represents the second highest value.

To further demonstrate the advantages of the developed method, we provide the receiver operating characteristic (ROC) curves of the test sequences used in Table 4, whose horizontal axis was the false-alarm ratio of the sequence and whose vertical axis was the probability of detection of the sequence. The ROC curves are illustrated in Figure 12. Note that again, we did not consider the SMSL and RIPT for the same reason. Tophat always performed the worst since the simple structural

elements could not handle the complex background. The performance of MPCM fluctuated greatly. For Sequences 1 and 2, MPCM worked very well but failed in the other sequences, which confirmed that MPCM is poor at addressing scenes filled with clutter and disturbances. IPI, NIPPS and ReWIPI achieved similar results, with the latter two performing slightly better than IPI on the whole. The biggest difference was in Sequence 3, which showed that ReWIPI performed the worst out of all methods. In general, the proposed method almost always reached the highest detection probability with respect to the same false-alarm ratio, implying that the proposed method outperformed the other state-of-the-art methods.

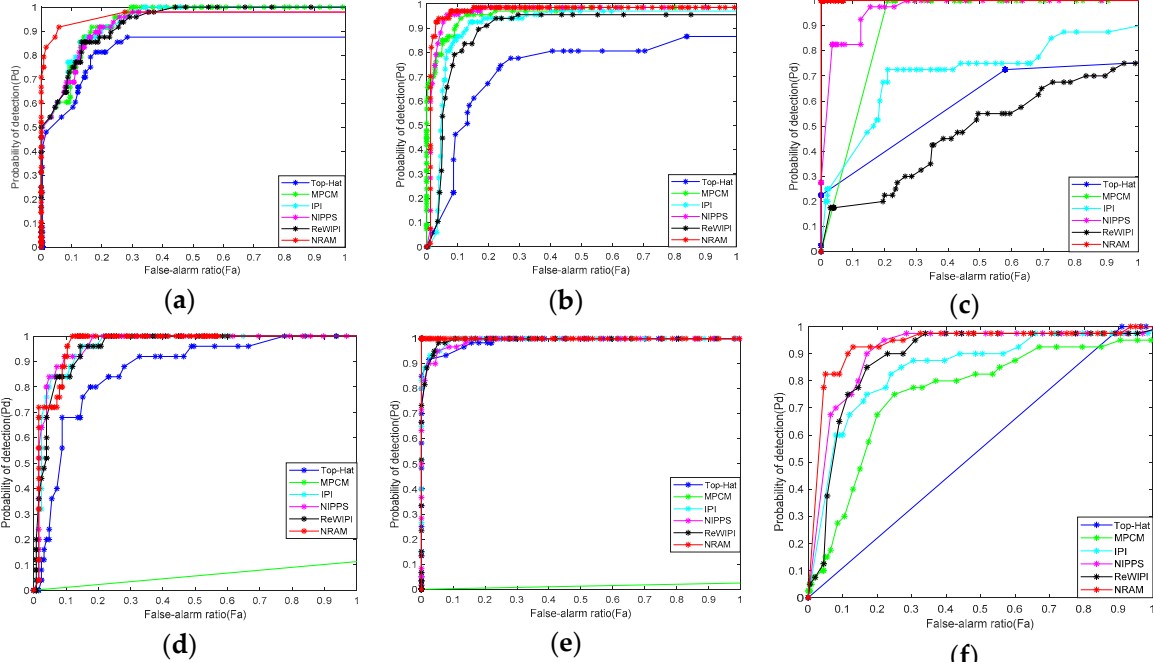

**Figure 12.** ROC curves of detection results of six real sequences. (**a**) Sequence 1; (**b**) Sequence 2; (**c**) Sequence 3; (**d**) Sequence 4; (**e**) Sequence 5; (**f**) Sequence 6.

Figure 13 displays the convergence rates of the last six methods. For the sake of comparison, the maximum value of relative error was set to 0.0008. Meanwhile, the condition of stopping iteration was that the relative error was less than $10^{-7}$ for all of the used methods. The IPI model and SMSL model both had slow convergence rates resulting from time-consuming APG, while the rest were based on the faster ADMM. Compared with IPI, the number of iterations in the proposed method was about 1/20 that of the IPI model. The RIPT model converged as fast as the NRAM, however, it had trouble dealing with the very dim target.

Moreover, to compare the computational cost of the nine methods more clearly, Table 5 shows the algorithm complexity and computing time for Figure 8B. Suppose the image size is and $m$, $n$ are the rows and columns of the patch-image. The computational cost of Tophat is $O(l^2 \log l^2)$, where the $l^2$ is the size of the structure element. Considering the image size, the final cost of Top-Hat is $O(l^2 \log l^2 MN)$. For LCM and MPCM, it is obvious that the major time-consuming part is calculating the saliency map pixel by pixel. A sliding window of size is needed for computing the saliency value of the central pixel. Thus, $k^2$ to $8k^2$ times mathematical operation per pixel is required, namely, in a single scale, the computational cost is $O(k^2 MN)$. Therefore, the total cost in all scales is $O(K^3 MN)$. For the rest of the methods, except for SMSL, the dominant factor is singular value decomposition (SVD), which has a computational complexity of $O(mn^2)$. Since an iterative reweighted scheme is used in ReWIPI, the cost of ReWIPI is $O(kmn^2)$. For SMSL, its complexity is determined by the subspace dimension and the matrix size, that is, $O(lmn)$, where $l$ denotes the subspace dimension.

It can be seen that the methods based on target and background separation (i.e., the last six methods) are generally slower than the filtering and HSV-based methods (i.e., the first three methods), resulting from the iterative procedure and singular value decomposition (SVD). However, considering the robustness and performance of each method, this sacrifice is acceptable. In addition, although SMSL and RIPT were faster than NRAM, their performances were not satisfactory in many cases. Therefore, there is no doubt that the developed NRAM model not only improves background suppression and target detection with complex backgrounds but also reduces the computational complexity compared with most baseline approaches.

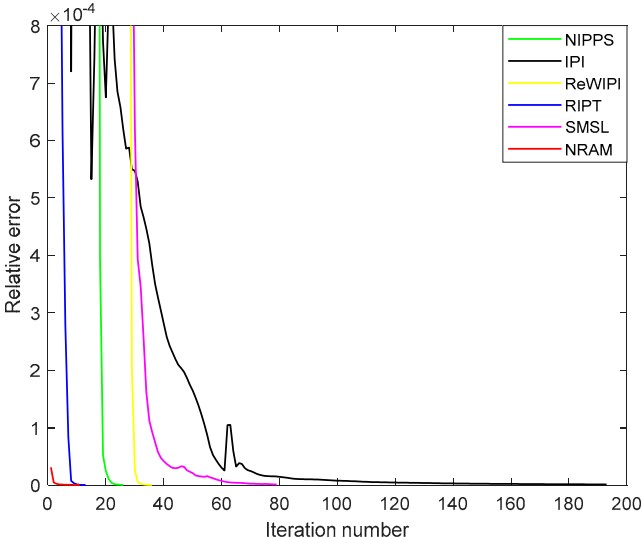

**Figure 13.** Convergence rates of the IPI, NIPPS, ReWIPI, SMSL, RIPT and NRAM models.

**Table 5.** Comparison of computational complexity and computing time of nine methods.

|  | Tophat | LCM | MPCM | IPI | NIPPS | ReWIPI | SMSL | RIPT | NRAM |
|---|---|---|---|---|---|---|---|---|---|
| Complexity | $O(I^2\log I^2 MN)$ | $O(K^3MN)$ | $O(K^3MN)$ | $O(mn^2)$ | $O(mn^2)$ | $O(kmn^2)$ | $O(lmn)$ | $O(mn^2)$ | $O(mn^2)$ |
| Time(s) | 0.022 | 0.074 | 0.089 | 11.907 | 7.486 | 15.469 | 1.245 | 1.352 | 3.378 |

## 5. Discussion

The separation of target and background utilizing optimization methods and sparse representation is widespread in the field of infrared small target detection based on the non-local self-correlation of infrared background and the sparsity of the target. The classic IPI exploits the nuclear norm to represent the low-rank patch-background matrix and when the infrared images are uniform, it works well. However, as discussed in Section 3, the larger the singular values, the farther the nuclear norm deviates. In fact, large singular values contain more details and information of the image. In other words, the nuclear norm cannot handle complex infrared scenes because of the rich details, which causes residuals and the reconstructed background is blurred. This is also the case with the $l_1$ norm. Unbalanced weights result in incomplete target reconstruction; that is the aforementioned over-shrink. NIPPS utilizes partial sum minimization of singular values (similar to the truncated nuclear norm) aiming to overcome the defect of the nuclear norm. One key step of NIPPS is to estimate the rank of the image, which is challenging. The small singular value of the image corresponds to the noise. Inaccurate estimation can easily minimize that part of the singular value, leading to the remaining components in the target image. ReWIPI and SMSL also make improvements to the nuclear norm and $l_1$ norm but still suffer from strong edges.

Currently, most state-of-the-art methods concentrate on the prior background and target, which is insufficient. RIPT introduces the structure tensor to address the problem of residuals. The performance

with complex scenes is much better but unfortunately, RIPT ignores low SNR situations which lack structural information, resulting in loss of the target. Based on the IPI model, to overcome the existing problems, a tighter surrogate of rank named the $\gamma$ norm with a weighted $l_1$ norm has been introduced to recover the background accurately. Motivated by the structural information of edges and remaining noise in the target images, the structured $l_{2,1}$ norm can handle the residuals well. Therefore, as well as the background and the target, a third component, which we called strong edges (actually not just edges), was developed. Namely, the $\gamma$ norm only needs to recover a low-rank component of the background, so the weighted $l_1$ norm only has to reconstruct the target and the $l_{2,1}$ norm is in charge of the components that are sparse but belong to the background. A single recovery task for each component makes the problem easier to solve.

There are several vital parameters in the proposed NRAM model including the penalty factor $\mu$, the norm factor $\gamma$ and the compromising constant *C*. As analyzed in Section 4, the $\gamma$ norm has some disadvantages but thanks to the penalty factor $\mu$, we can overcome them. The norm factor $\gamma$, affecting how close the surrogate is to the true rank and the compromising constant *C*, is the key factor determining the recovery of the target image. We have discussed experiments around choosing the best value of them with self-adaption.

By comparing the performance of the proposed method with eight state-of-the-art methods from qualitative and quantitative perspectives, it is fair to say that the proposed NRAM model showed relative superiority in both background suppression and target enhancement. Furthermore, the robustness of the algorithm when faced with various scenes outperformed the baselines.

## 6. Conclusions

To improve the ability of background suppression with complex backgrounds, a novel method based on non-convex rank approximation minimization was proposed to reconstruct the low-rank component more accurately, namely the background. Furthermore, a more powerful sparse constraint item, the weighted $l_1$ norm, was exploited to enhance the target detection performance. Considering the common challenge that most state-of-the-art methods currently face, which is an inability to completely wipe out the strong edges and rare structures, an additional structured $l_{2,1}$ norm was employed to address this issue. By carefully choosing the vital parameters, an optimization algorithm combining ADMM and DC strategy was presented to solve this new model. Experimental results based on diverse scenes and sequences demonstrated that the proposed method outperformed the other state-of-the-art methods, showing more robustness and less computational complexity.

Several problems still need to be solved. For instance, if there are non-target elements at one of the four corners of the image (i.e., lower-left corner) that are much more salient than the target, all methods including the proposed NRAM model would incorrectly regard the non-target noise as the real target. Our plan is to utilize more features of targets to depict the target component better. Moreover, seeking the major dissimilarity between non-target components and targets also plays an important role. In addition, although our method reduced the computing time when compared with similar methods, it was still inferior to the traditional filtering method, which is a problem that needs to be solved.

**Author Contributions:** L.Z. proposed the original idea, performed the experiments and wrote the manuscript. L.P., T.Z., S.C. and Z.P. contributed to the writing, direction and content and revised the manuscript.

**Funding:** This work was funded by National Natural Science Foundation of China (61571096, 61775030 and 61575038) and the Key Laboratory Fund of Beam Control, Chinese Academy of Sciences (2017LBC003).

**Acknowledgments:** The authors would thank the published code of Gao's model and Dai's model for comparison and Xiaoyang Wang who kindly provided images and the code of one compared model.

**Conflicts of Interest:** The authors declare no conflict of interest.

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
