# Peer review of "Infrared Small Target Detection via Non-Convex Rank Approximation Minimization Joint l2,1 Norm"

_remotesensing, doi:10.3390/rs10111821_

Round 1
Reviewer 1 Report
p { margin-bottom: 0.08in; }This paper proposes a method for small target enhancement with application to infrared images. The method is formulated as finding the best decomposition of the image into their background, target and noise components. The decomposition is computed from the minimization of an objective function that combines the gamma-norm of the background image, the Lw-1 norm of the target, and the L1-1 norm of the edges of the images. With respect to other state of the art methods, the differences of the proposed objective functional seem to outperform these methods in some difficult practical cases. The objective function ia a non-convex problem. It is nicely approached using the ADMM method combined with difference convex programming (DC) in the optimization of the non-convex subproblem. The authors show an extensive evaluation of the proposed method in different difficult scenarios comparing with the state-of-the-art most related to their work. The method usually outperfoms the state-of-the-art, although the authors show some failure cases.
This paper deals with a very difficult problem in remote sensing: the detection of small targets in infrared images. The proposed method formulates the problem with an original model and provides the derivations of the equations that may be interesting for other problems based on ADMM in non-convex optimization problems. I believe that this work is more exciting from the methodological point of view than from the application side which is something to acknowledge.
The paper is very well written and the experimental section is thorough and accurately discussed. So I do not have any specific comment to improve the manuscript. Maybe, it would be very interesting to add a little discussion on the use of ADMM in non-convex objective functions arising in computer vision applications.
Author Response
Point 1: The paper is very well written and the experimental section is thorough and accurately discussed. So I do not have any specific comment to improve the manuscript. Maybe, it would be very interesting to add a little discussion on the use of ADMM in non-convex objective functions arising in computer vision applications.
Response 1: I am very grateful to your comments for the manuscript. Your suggestion is very useful for enriching the introduction of the paper. According with your advice, we amended the relevant part in manuscript. Of course, it’s important to introduce ADMM before using this method. So, in the Introduction part, we gave a brief introduction to ADMM in section 1 to show its superiority, especially in the field of computer vision and image processing. Moreover, all the applications are related to non-convex problems. And the manuscript is now well edited.
The specific additions are as follows:
Furthermore, alternating direction method of multipliers (ADMM) [31] has a faster convergence rate when compared with APG, and its precision is also higher [32]. Hence, ADMM is a popular method to solve optimization problems in many fields, particularly in the computer vision and image processing field. For instance, Gu et al. [33] utilized ADMM to solve the non-convex weighted nuclear norm minimization (WNNM) problem efficiently, which was successfully applied to image inpainting. Xue et al. [34] proposed a non-convex low-rank tensor completion model using ADMM to obtain the best recovery result in color images. Based on total variation regularized tensor RPCA, Cao et al. [35] designed a non-convex and non-separable model to complete background subtraction with an ADMM solver.

Reviewer 2 Report
1. The English of this paper have to be improved. Please send it to a native speaker to make corrections.
2. In Fig. 6, you have to increase the size of figures, and to remove some of them that may are not needed. It seems that there exists plots with the same parameters.
3. Add also the computational cost of the proposed method.
4. Please, provide a webpage (in the paper) that will include the used dataset and your results for people that are interesting for comparisons.
5. In addition, in order to improve your introduction (related work) you have to cite the following related works.
[1] Benedek, C., Descombes, X., Zerubia, J., 2012. Building development monitoring in multitemporal remotely sensed image pairs with stochastic birth-death dynamics. IEEE Trans. Pattern Anal. Mach. Intell. 34, 33–50.
[2] Liu, D., Li, Z., Liu, B., Chen, W., Liu, T., & Cao, L. (2017). Infrared small target detection in heavy sky scene clutter based on sparse representation. Infrared physics & technology, 85, 13-31.
[3] Zhang, H., Bai, J., Li, Z., Liu, Y., & Liu, K. (2017). Scale invariant SURF detector and automatic clustering segmentation for infrared small targets detection. Infrared physics & technology, 83, 7-16.
[4] Li, Z., Zhang, H., Bai, J., Zhou, Z., & Zheng, H. (2018, April). A speeded-up saliency region-based contrast detection method for small targets. In Ninth International Conference on Graphic and Image Processing (ICGIP 2017) (Vol. 10615, p. 1061506). International Society for Optics and Photonics.
Author Response
Point 1: The English of this paper have to be improved. Please send it to a native speaker to make corrections.
Response 1: Thank you very much for your review report. These comments are all valuable and very helpful for revising and improving our paper, as well as the important guiding significance to our researches. We have studied comments carefully and have made correction which we hope meet with approval. And now, the manuscript is well edited.
Point 2: In Fig. 6, you have to increase the size of figures, and to remove some of them that may are not needed. It seems that there exists plots with the same parameters.
Response 2: Thank you for pointing this out. Fig. 6 and the font inside have been enlarged. Moreover, we have not use black in the last two parameters’ ROC curves for the sake of visualization. For each parameter, five figures correspond to five sequences. And there are five or six variations per parameter, so, all the figures in Fig. 6 are needed. For the last figure, the reason why the ROC curves look like the same is that the target in sequence 5 is easy to detect while the non-target noise is hard to suppress (details in section 4.3.3). (Enlarged figure is in the Word file)
Point 3: Add also the computational cost of the proposed method.
Response 3: Thank you for pointing this out. Computational cost comparison and related explanations have been added in section 4.5.
The specific additions are as follows:
Moreover, to compare the computational cost of the nine methods more clearly, Table 5 shows the algorithm complexity and computing time for Figure 8B. Suppose the image size is M×N, and m, n are the rows and columns of the patch-image. The computational cost of Tophat is O(I2logI2), where the I2 is the size of the structure element. Considering the image size, the final cost of Top-Hat is O(I2logI2MN). For LCM and MPCM, it is obvious that the major time-consuming part is calculating the saliency map pixel by pixel. A sliding window of size k×k (k=1,2,...K) is needed for computing the saliency value of the central pixel. Thus, k2 to 8k2 times mathematical operation per pixel is required, namely, in a single scale, the computational cost is O(k2MN). Therefore, the total cost in all scales is O(K3MN). For the rest of the methods, except for SMSL, the dominant factor is singular value decomposition (SVD), which has a computational complexity of O(mn2). Since an iterative reweighted scheme is used in ReWIPI, the cost of ReWIPI is O(kmn2). For SMSL, its complexity is determined by the subspace dimension and the matrix size, i.e. O(lmn), where l denotes the subspace dimension.
It can be seen that the methods based on target and background separation (i.e., the last six methods) are generally slower than the filtering and HSV-based methods (i.e., the first three methods), resulting from the iterative procedure and singular value decomposition (SVD). However, considering the robustness and performance of each method, this sacrifice is acceptable. In addition, although SMSL and RIPT were faster than NRAM, their performances were not satisfactory in many cases. Therefore, there is no doubt that the developed NRAM model not only improves background suppression and target detection with complex backgrounds, but also reduces the computational complexity compared with most baseline approaches. (Table 5 is in the Word file)
Point 4: Please, provide a webpage (in the paper) that will include the used dataset and your results for people that are interesting for comparisons.
Response 4: Thank you for your suggestion. We have considered this before; however, the dataset used in this paper is not an open dataset. In fact, to the best of our knowledge, there are no open datasets in the field of infrared small target detection because of some possible military factors. We will offer the Matlab code (the GitHub link) for the people that are interested for comparisons. This is the Github link: https://github.com/Lanneeee/NRAM.
Point 5: In addition, in order to improve your introduction (related work) you have to cite the following related works.
[1] Benedek, C., Descombes, X., Zerubia, J., 2012. Building development monitoring in multitemporal remotely sensed image pairs with stochastic birth-death dynamics. IEEE Trans. Pattern Anal. Mach. Intell. 34, 33–50.
[2] Liu, D., Li, Z., Liu, B., Chen, W., Liu, T., & Cao, L. (2017). Infrared small target detection in heavy sky scene clutter based on sparse representation. Infrared Physics & Technology, 85, 13-31.
[3] Zhang, H., Bai, J., Li, Z., Liu, Y., & Liu, K. (2017). Scale invariant SURF detector and automatic clustering segmentation for infrared small targets detection. Infrared physics & technology, 83, 7-16.
[4] Li, Z., Zhang, H., Bai, J., Zhou, Z., & Zheng, H. (2018, April). A speeded-up saliency region-based contrast detection method for small targets. In Ninth International Conference on Graphic and Image Processing (ICGIP 2017) (Vol. 10615, p. 1061506). International Society for Optics and Photonics.
Response 5: Thank you for pointing out the inadequacy of the research background (introduction and related work) in the paper; we have added more relevant references (including the above four) in section 1 and section 2.

Reviewer 3 Report
This article describes a non-convex rank approximation minimization (NRAM) model. The results show that the proposed model is better than other competing models.
I find the paper to be interesting; the testing procedure is well planned and well described. The contribution seems specified in the “Related works” part and I consider it a valuable contribution to the Remote Sensing journal.
Some specific concerns are as follows:
It is difficult to appreciate in Figure 6 the parameter values of de ROC curves. The authors should improve their visualization. The same goes for the graphic presentation of Figure 11.
Author Response
Point 1: It is difficult to appreciate in Figure 6 the parameter values of the ROC curves. The authors should improve their visualization. The same goes for the graphic presentation of Figure 11.
Response 1: I am very grateful to your comments for the manuscript. Your comments are all valuable and very helpful for revising and improving our paper. According with your advice, we amend the relevant part in manuscript. We have enlarged the size of Figure 6 and Figure 11 to improve their visualization. (Enlarged figures are in the Word file)
